# Unraveling the role of urea hydrolysis in salt stress response during seed germination and seedling growth in *Arabidopsis thaliana*

Yuanyuan Bu[1,2]*, Xingye Dong[1,2], Rongrong Zhang[1,2], Xianglian Shen[1,2], Yan Liu[1,2], Shu Wang[3], Tetsuo Takano[4], Shenkui Liu[3]*

[1]Key Laboratory of Saline-Alkali Vegetation Ecology Restoration, Northeast Forestry University, Ministry of Education, Harbin, China; [2]College of Life Sciences, Northeast Forestry University, Harbin, China; [3]State Key Laboratory of Subtropical Silviculture, Zhejiang A&F University, Hangzhou, China; [4]Asian Natural Environmental Science Center (ASNESC), University of Tokyo, Tokyo, Japan

**Abstract** Urea is intensively utilized as a nitrogen fertilizer in agriculture, originating either from root uptake or from catabolism of arginine by arginase. Despite its extensive use, the underlying physiological mechanisms of urea, particularly its adverse effects on seed germination and seedling growth under salt stress, remain unclear. In this study, we demonstrate that salt stress induces excessive hydrolysis of arginine-derived urea, leading to an increase in cytoplasmic pH within seed radical cells, which, in turn, triggers salt-induced inhibition of seed germination (SISG) and hampers seedling growth. Our findings challenge the long-held belief that ammonium accumulation and toxicity are the primary causes of SISG, offering a novel perspective on the mechanism underlying these processes. This study provides significant insights into the physiological impact of urea hydrolysis under salt stress, contributing to a better understanding of SISG.

**\*For correspondence:**
yuanyuanbu@nefu.edu.cn (YB);
shenkuiliu@nefu.edu.cn (SL)

## Editor's evaluation

This important study advances our understanding of the molecular mechanism underlying the inhibition of seed germination and seedling growth by salt stress. The evidence supporting the conclusions is convincing, with rigorous genetic, physiological, and metabolic analyses. This paper will be of interest to plant biologists and crop breeders.

## Introduction

In an era marked by global climate change, land degradation, and biodiversity loss, ensuring food security for a world population expected to exceed 9.7 billion by 2050 presents a formidable challenge (*Majeed et al., 2023*). Currently, to boost plant productivity, agricultural products primarily use synthetic chemical fertilizers, which, despite their benefits, have negative consequences on the environment and soil biodiversity (*Hu et al., 2022*). Fertilizers, particularly those containing urea due to its high water solubility, play a crucial role in enhancing soil fertility. Urea, a key synthetic nitrogen source, constitutes about half of the nitrogen applied to crop production globally and is the most widely used nitrogen fertilizer in agriculture (http://faostat.fao.org). However, the adverse effects of urea-based fertilizers on seed germination and seedling growth are still not reasonably explained (*Bremner and Krogmeier, 1988*; *Bremner and Krogmeier, 1989*; *Bu et al., 2015*). Therefore, a

**Figure 1.** Blocking arginine hydrolysis and promoting seed germination under salt stress. (**a**) A simple model of arginine metabolism in *Arabidopsis thaliana*. This model outlines the conversion of arginine into (1) nitric oxide (NO) and citrulline by nitric oxide synthase (NOS); (2) polyamine (PA) by arginine decarboxylase (ADC); and (3) ornithine and urea by arginase, with urea further decomposed to ammonia by urease. It highlights arginine as a shared competitive substrate for the three enzymes ARGAH, NOS, and ADC, illustrating the competitive enzymatic interactions. NOHA: $N^G$-hydroxy-L-arginine, an arginase inhibitor; PPD: phenyl phosphorodiamide, a urease inhibitor. (**b**) Comparison of germination rates of WT seeds on half-strength MS (½ MS) medium containing 0 mM (control) or 135 mM NaCl with 5 µM NOHA or without NOHA. Photographs were taken at the 48 hr after 2 days at 4°C. A representative result from one of three independent experiments, all yielding similar outcomes, is shown. Scale bar represents 1 mm. (**c**) Seeds were germinated on ½ MS medium containing either 0 mM or 135 mM NaCl and varying concentrations of NOHA (0, 1.5, 2.5, 5 µM); photographs were taken 14 days after germination. (**d**) Germination rates of WT seeds in either 0 mM or 135 mM NaCl medium with or without 5 µM NOHA or 15 µM PPD. Stratification consisted of pretreatment of seeds for 2 days at 4°C in the darkness. In all experiments, seeds were freshly sowed and incubated under 16 hr light and 8 hr dark conditions at 22°C. The experiment was repeated three times; at least 30 seeds were counted in each replicate. The data was analyzed using a one-way ANOVA, followed by Duncan's multiple range test for the post hoc comparisons. Significant differences between groups, indicated by different letters on the error bars, were determined ($p<0.05$).

The online version of this article includes the following figure supplement(s) for figure 1:

**Figure supplement 1.** Effects of $N^G$-hydroxy-L-arginine (NOHA) on WT seedling growth.

deeper understanding of urea's hydrolysis, transport, and utilization within plants is imperative for devising informed strategies aimed at crop improvement.

Urea serves not only the primary nitrogen source that plants actively absorb from the soil solution but also an intermediate in plant arginine catabolism, which is involved in the nitrogen remobilization of nitrogen from source tissues. Arginase (ARGAH; arginine amidinohydrolase), the only enzyme in plants known to generate urea in vivo, plays a pivotal role in the mobilization of seed storage reserves during early seedling growth post-germination. The catabolism of arginine by arginase is crucial for mobilizing stored nitrogen upon germination and redistributing nitrogen from source tissues, as evidenced by various studies (*Goldraij and Polacco, 1999*; *Todd et al., 2001*; *Todd and Gifford, 2002*). Arginine, abundant in the storage proteins of most plant seeds, has the highest nitrogen-to-carbon ratio (N:C = 4:6) among all amino acids, making it a significant nitrogen source for seed reserve mobilization (*Jones and Boulter, 1968*; *King and Gifford, 1997*; *Polacco et al., 2013*). In a study of 379 angiosperm seeds, arginine accounted for 17% of the seed nitrogen content (8.58 g arginine/16 g seed N) (*Van Etten et al., 1967*).

Beyond its hydrolysis by arginase to yield urea, arginine also serves as a substrate for nitric oxide (NO) and polyamine (PA) biosynthesis in plants by nitric oxide synthase (NOS) and arginine

decarboxylase (ADC) (*Siddappa and Marathe, 2020*), respectively. Consequently, arginine becomes a shared substrate, competitively utilized by the three enzymes ARGAH, NOS, and ADC (*Figure 1*). Many recent studies suggested that the improved salt tolerance observed in arginase-deficient mutants is attributed to an indirect upregulation of NOS and ADC pathways, thereby enhancing NO and PA-mediated plant defense responses (*Flores et al., 2008*; *Shi and Chan, 2013a*; *She et al., 2017*; *Wang et al., 2011*; *Winter et al., 2015*). However, this raises the question of how the presence of arginase activity still allows urea-deficient mutants to significantly alleviate salt-induced inhibition of seed germination (SISG) and bolster seedling growth (*Bu et al., 2015*).

To address these questions, our study focused on the connection between urea produced by arginase hydrolysis pathway, distinct from the PA and NO pathways, and the SISG and seedling growth, with a particular focus on the dual-step hydrolysis of arginine (*Figure 1*) and the internal transport of urea in relation to SISG and seedling growth. By employing specific enzyme inhibitors and gene deletion mutants within the arginine two-step hydrolysis pathway, we found that hydrolysis of arginine-derived urea is the key point for effectively alleviating the adverse effects of salt on seed germination and seedling growth. This led us to the question: what mechanism underlies the triggering of SISG and stunted seedling growth by urea hydrolysis? We ruled out the potential accumulation of ammonium, resulting from urea hydrolysis, by incorporating exogenous ammonium into the salt stress medium. Interestingly, our findings on intracellular pH measurements indicate that the salt-induced hydrolysis of urea, yielding $OH^-$, leads to an increase in intracellular pH of the radicle, pinpointing this as the primary factor initiating SISG and impeding seedling growth. This finding challenges our previous hypothesis, proposed by *Bu et al., 2015*, that $NH_4^+$ accumulation resulting from urea hydrolysis under salt stress is the trigger for SISG. These new insights provide a novel perspective of mechanisms driving SISG and subsequent seedling development.

## Results

### Urea hydrolysis is the cause of SISG in the two-step arginine hydrolysis

Initially, to determine whether arginase-mediated arginine hydrolysis pathway is involved in SISG and seedling growth, we utilized two inhibitors specific to the two-step hydrolysis reaction of the arginine hydrolysis pathway: $N^G$-hydroxy-L-arginine (NOHA), an arginase inhibitor, and phenyl phosphorodiamidate (PPD), a urease inhibitor (*Figure 1a*). We analyzed the seed germination phenotypes under these conditions. In the absence of NOHA, the NaCl treatment significantly hindered the seeds' capacity to germinate, as evidenced by the radicle's inability to break through the seed coat (*Figure 1b*). However, the addition of 5 µM NOHA into NaCl medium significantly increased the germination rate (*Figure 1b*). In contrast, the presence or absence of NOHA in the control medium did not notably affect seed germination (*Figure 1b*).

To better observe the effect of NOHA on seed germination or early seedling growth, different concentrations of NOHA (0, 1.5, 2.5, and 5 µM) were added to the control or NaCl mediums (*Figure 1c*). The results showed that the inhibitory effect of salt on seed germination was significantly alleviated by the increasing NOHA concentrations, while NOHA added in control medium had no substantial effect (*Figure 1c*). Additionally, root length measurements taken 14 days after germination indicated that NOHA could promote root growth under salt stress conditions (*Figure 1—figure supplement 1*).

To directly elucidate the impact of the arginase-mediated arginine hydrolysis pathway on SISG, and under the assumption that the competition for arginine substrate by multiple pathways (*Figure 1*) is not affected at this stage, we investigated the effect of urea hydrolase inhibitor PPD, which acts on a downstream metabolite of arginine, on seed germination. This was then compared to the effects observed in the NOHA assay. The results showed that salt significantly inhibited the germination of wild-type *Arabidopsis* seeds, with less than 10% germination. The addition of 15 µM of PPD increased the germination rate to 70%, compared with the rate without PPD (*Figure 1d*), which showed the same trend as NOHA treatment (*Figure 1d*). These results indicate that salt tolerance in WT seeds germination and seedling growth is manifested especially under NOHA and PPD conditions. The two-step hydrolysis pathway of arginine mediated by arginase and its downstream metabolite urea hydrolase is not only involved in SISG events, but also urea hydrolysis may be the leading cause of SISG occurrence.

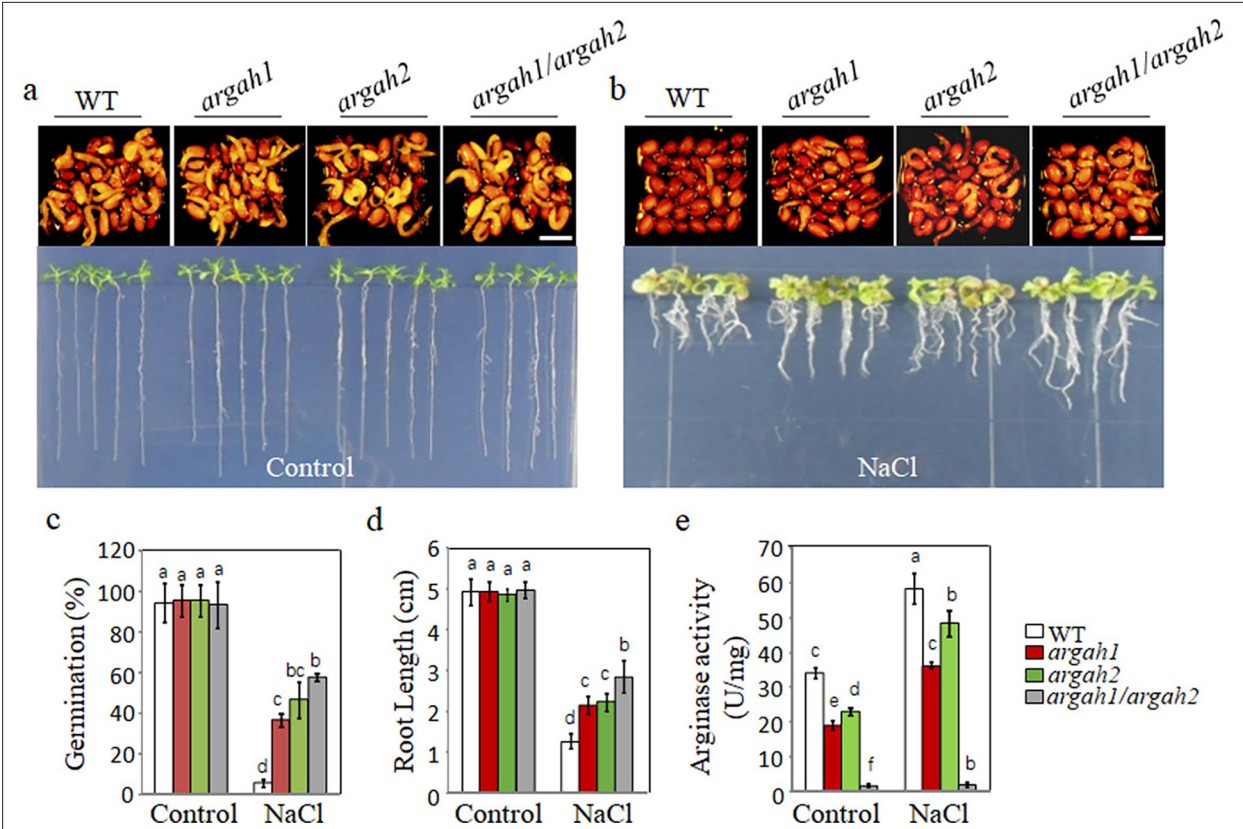

**Figure 2.** Deletion of arginase hydrolytic pathway alleviates the inhibition of seed germination by salt. WT, *argah1*, *argah2*, and *argah1/argah2* mutant seeds germinated, and seedling growth was observed on ½ MS medium under two conditions: 0 mM NaCl (control) (**a**) and 135 mM NaCl (**b**). Seedlings images of seeds germination 48 hr after 2 days at 4°C and 14 days after germination. (**c**) Germination rates, (**d**) root length measurements, and (**e**) arginase activity assays for WT, *argah1*, *argah2*, and *argah1/argah2* mutants on control and 135 mM NaCl mediums. In all experiments, seeds were freshly sowed and incubated under 16 hr light and 8 hr dark conditions at 22°C. The experiment was repeated three times; at least 30 seeds were counted in each replicate. Scale bar: 1 mm. Data were subjected to a one-way ANOVA, followed by Duncan's post hoc test. Different letters on the error bars denote significant differences in the data ($p<0.05$).

The online version of this article includes the following source data and figure supplement(s) for figure 2:

**Figure supplement 1.** Characterization of *atargah1*, *atargah2,* and the *atargah1/atargah2* double mutant plants.

**Figure supplement 1—source data 1.** Sequencing results of the *AtArgAH1/AtArgAH2* double homozygous mutant lines *atargah1/atargah2*.

## Genetic evidence for SISG triggered by the arginine hydrolysis pathway

To clarify the genetic basis of arginase inhibitor experiments described above, the following studies were conducted. *Arabidopsis thaliana* has two genes encoding ARGAH, namely ARGAH1 (gene number: AT4G08900) and ARGAH2 (gene number: AT4G08870). *AtArgAH1* has a T-DNA insertion within its first exon, while *AtArgAH2* contains a T-DNA insertion in the fourth exon. Consequently, the mutants resulting from these insertions were denoted as *argah1* and *argah2*, respectively (*Figure 2—figure supplement 1a*). Additionally, *AtArgAH1* and *AtArgAH2* double knockout mutants (*argah1/argah2*) were generated using clustered regularly interspaced short palindromic repeats (CRISPR)/ of CRISPR-associated (Cas) protein 9 system. Targeted modifications were introduced into the third exon of the *AtArgAH1* gene and the second exon of the *AtArgAH2* gene. In total, four hygromycin-resistant lines were obtained from the $T_0$ transgenic lines. Subsequent to CRISPR-Cas9 gene editing and sequencing, one line of the *argah1/argah2* double mutant exhibited a 1 bp insertions in the coding region of the *AtArgAH1* gene, and another 1 bp insertions in the coding region of the *AtArgAH2* gene (*Figure 2—figure supplement 1b*). Thus, for subsequent genetic phenotypic analyses, *argah1/argah2* double mutants were utilized alongside wild-type *Arabidopsis* (WT). Seed germination and initial growth after germination of all mutants (*argah1*, *argah2*, and *argah1/argah2*) were not significantly

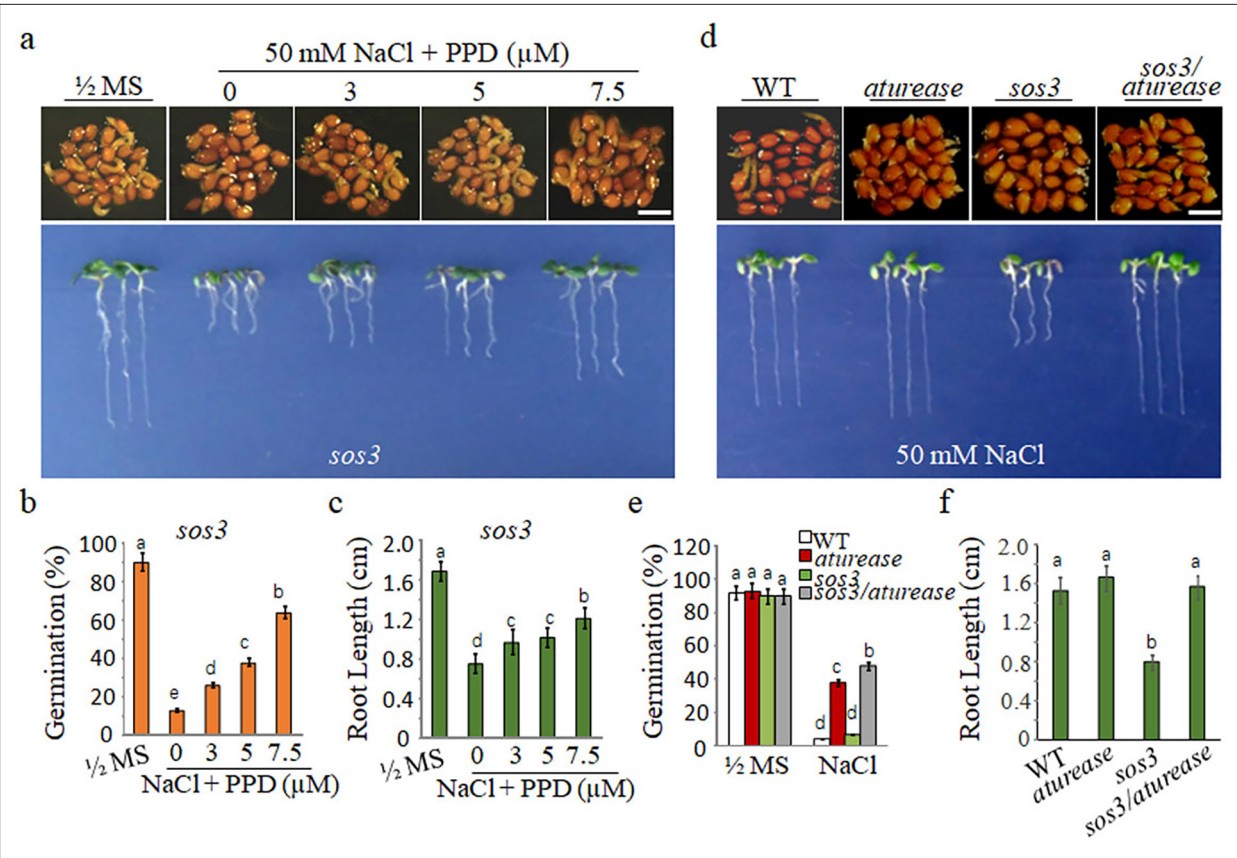

**Figure 3.** Blocking the arginine hydrolysis pathway mitigates the hypersensitivity of *sos3* mutants to salt. (**a**) Seeds germinated and seedling growth of *sos3* mutants on ½ MS with 50 mM NaCl medium, supplemented with phenyl phosphorodiamidate (PPD) at different concentrations (0, 3, 5, and 7.5 μM). Seedlings images of seeds germination 48 hr after 2 days at 4°C and 14 days after germination. (**b**) Germination rates and (**c**) root lengths of *sos3* mutants assessed after the specified treatments. (**d**) WT, *aturease*, *sos3*, and *sos3/aturease* seeds germinated and grew on ½ MS with 50 mM NaCl. Seedlings images of seeds germination 48 hr after 2 days at 4°C and 14 days after germination. (**e**) Germination rate and (**f**) root lengths for WT, *sos3*, *aturease,* and *sos3/aturease* mutants were measured after the indicated treatments. In all experiments, seeds were freshly sowed and incubated under a 16 hr light/8 hr dark cycle at 22°C. The experiment was repeated three times; at least 25 seeds were counted in each replicate. Scale bar: 1 mm. The data were analyzed using one-way ANOVA followed by Duncan's post hoc test, with different letters on the error bars indicating significant differences in the data (p<0.05).

different from those of WT in the control medium (***Figure 2a***). In contrast, adding NaCl to the control medium severely inhibited seed germination and radicle development of WT (***Figure 2b***). The seed germination rates of each mutant (*argah1, argah2*, and *argah1/argah2*) were significantly higher than those of WT under NaCl conditions (***Figure 2c***). Furthermore, the root length of the mutant under salt stress was significantly higher than that of the WT (***Figure 2d***). Moreover, arginase activity analysis revealed significantly lower levels in *argah1, argah2*, and *argah1/argah2* mutants following NaCl treatment compared to WT (***Figure 2e***). These findings indicate that partial or complete blocking of the arginine hydrolysis pathway can effectively alleviate SISG. We have previously reported that deletion mutants of the urea hydrolase gene (*urease*) enhance salt tolerance during seed germination (***Bu et al., 2015***). That implies that even the accumulation of urea in vivo, derived from the hydrolysis of arginine by arginase, is insufficient to trigger SISG.

It is well known that Na+ toxicity is the main factor triggering SISG. The mechanism involves SOS3 binds to free Ca2+, consequently activating SOS2 protein kinase, which in turn phosphorylates SOS1, leading to the activation of SOS1 transport responsible for pumping Na+ out of the cell (***Zhou et al., 2022***). Therefore, SOS3-deficient mutant (*sos3*) accumulate substantial Na+ levels and are highly sensitive to salt stress. In order to further clarify the importance of urea hydrolysis in SISG, *sos3* mutants were considered as standard materials for SISG analysis, along with *aturease/sos3* double mutant. Initially, we introduced PPD to 50 mM NaCl medium to observe the growth of *sos3*. Notably, with

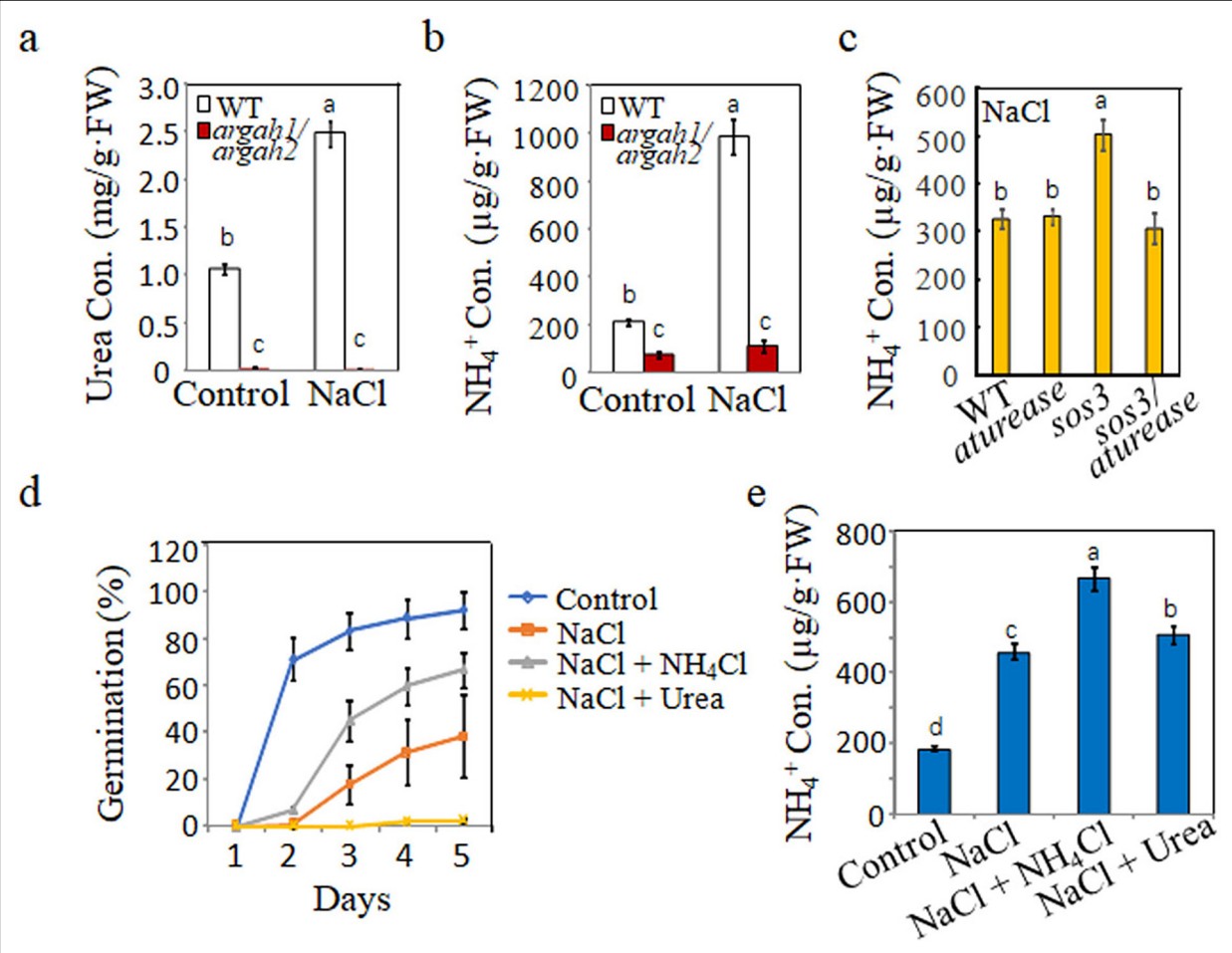

**Figure 4.** Correlation between metabolites from arginine hydrolysis and salt inhibit-induced inhibition of seed germination. (**a**) Urea and (**b**) NH$_4^+$ concentrations were measured in WT and *argah1/argah2* seedlings grown in ½ MS medium under control (0 mM) and 135 mM NaCl conditions. (**c**) NH$_4^+$ levels assessed in WT, *aturease*, *sos3*, and *sos3/aturease* seedlings under 50 mM NaCl treatment in ½ MS. (**d**) Effect of NH$_4$Cl on salt response was evaluated in WT seeds, monitoring germination rates in control medium with 135 mM NaCl, with and without 10 mM NH$_4$Cl and 10 mM urea, over time. (**e**) NH$_4^+$ concentration in WT seedlings measured on ½ MS with 135 mM NaCl, with or without the addition of 10 mM NH$_4$Cl and 10 mM urea. In all experiments, seeds were freshly sowed and incubated under a 16 hr light/8 hr dark cycle at 22°C. The experiment was repeated three times; at least 30 seeds were counted in each replicate. Scale bar: 1 mm. The data were analyzed using a one-way ANOVA, with Duncan's test for post hoc comparison. Different letters on the error bars indicate significant differences in the data (p<0.05).

The online version of this article includes the following figure supplement(s) for figure 4:

**Figure supplement 1.** Effect of NH$_4^+$ or urea on NaCl responses of wild-type (WT) germinated seeds.

increasing the concentrations of PPD, the growth inhibition of *sos3* induced by NaCl was alleviated significantly (*Figure 3a*). Furthermore, both seed germination rate (*Figure 3b*) and 14-day root length (*Figure 3c*) were significantly increased compared to those without PPD treatment. Subsequently, the *aturease/sos3* double mutant analysis revealed a significant improvement in germination ability and growth compared to *sos3* in the presence of 50 mM NaCl (*Figure 3d and e*). In addition, the root length of *aturease/sos3* double mutant exceeded that of *sos3* mutants under salt stress conditions (*Figure 3f*). The above treatments with urea hydrolase inhibitors PPD, coupled with genetic evidence, suggest that blocking the hydrolysis of arginine-derived urea can mitigate the hypersensitivity of *sos3* to salt stress. This further corroborates the fact that urea accumulation in vivo is not the primary trigger of SISG. Instead, urea hydrolysis appears to play a dominant role in this process.

## How does the hydrolysis of arginine-derived urea trigger SISG?

Next, if urea accumulation is not the key to SISG, is urea hydrolysis detrimental to seed germination? The cause of SISG by excessive urea hydrolysis in triggering SISG remains an enigmatic issue. Initially,

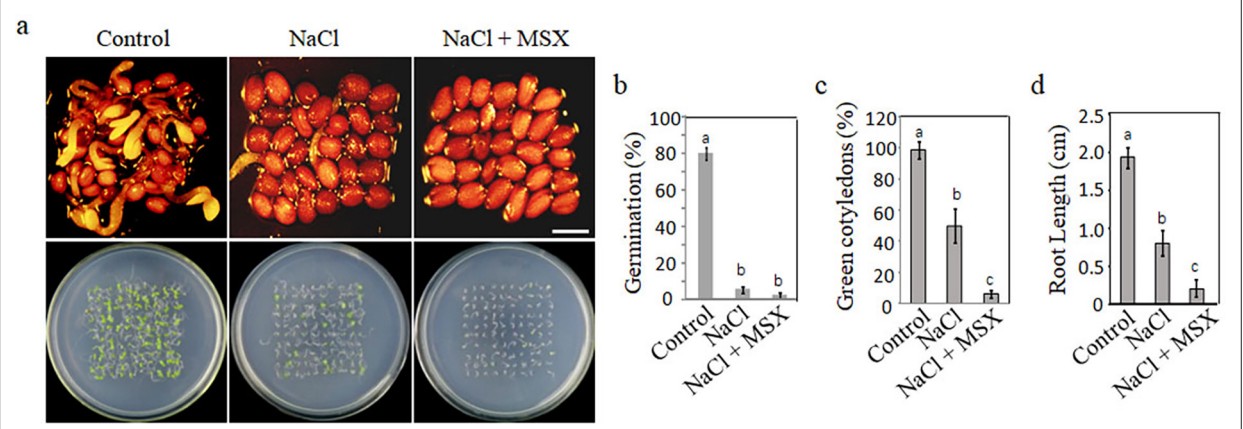

**Figure 5.** Effect of acidification process of glutamine synthetase assimilation on plant salt tolerance. (**a**) Germination of WT seeds on ½ MS medium (control), ½ MS medium supplemented with 135 mM NaCl, and ½ MS medium containing 3 µM L-methionine sulfoximine (MSX). Seedlings images of seeds germination 48 hr after 2 days at 4°C and 10 days after germination. Scale bar represents 1 mm. The study evaluated (**b**) WT seeds germination rates, (**c**) the count of green cotyledons, and (**d**) root length. In all experiments, seeds were freshly sowed and incubated under 16 hr light and 8 hr dark conditions at 22°C. The experiment was repeated in triplicate with a minimum of 30 seeds per replicate. The data analysis was performed using a one-way ANOVA and Duncan's post hoc test, with significant differences ($p < 0.05$) denoted by different letters on the error bars.

we confirmed that NaCl stress resulted in the accumulation of urea (*Figure 4a*) and $NH_4^+$ (*Figure 4b*) during WT seed germination, suggesting that salt stress promoted seed nitrogen mobilization and significantly lowered urea and $NH_4^+$ levels in the *argah1/argah2* double mutants (*Figure 4a and b*). In addition, in the presence of 50 mM NaCl, *sos3*-mutant seeds accumulated higher $NH_4^+$ levels than WT and *aturease* mutants. Remarkably, the *aturease/sos3* double mutant effectively reduced $NH_4^+$ levels to those akin to WT and *aturease* (*Figure 4c*). These findings initially indicated that SISG might be triggered by the ammonium accumulation resulting from excessive hydrolysis of arginine-derived urea. However, the following experiments challenged this seemingly reasonable hypothesis. We examined seed germination and growth in the presence of both NaCl and $NH_4Cl$. Surprisingly, exogenous $NH_4^+$ significantly alleviated the inhibitory effects of NaCl on seed germination (*Figure 4d*, *Figure 4—figure supplement 1*), with a concomitant increase in $NH_4^+$ (*Figure 4e*), indicating that $NH_4^+$ accumulation was not the primary cause of SISG. Further validation was sought by adding urea to NaCl medium. Intriguingly, urea substantially inhibited seed germination (*Figure 4d*, *Figure 4—figure supplement 1*), but $NH_4^+$ content did not accumulate in seeds (*Figure 4e*), thereby confirming that $NH_4^+$ produced via urea hydrolysis was not the driving force behind SISG.

These results disproved that $NH_4^+$ accumulation and toxicity caused by urea hydrolysis was the primary cause of SISG. Then, we reconsidered the hydrolysis process of urea, based on the complete reaction of urea hydrolysis: $(NH_2)_2CO + 3H_2O \rightarrow 2NH_4^+ + HCO_3^- + OH^-$, and discovered that in addition to $NH_4^+$, excessive urea hydrolysis by alkaline reaction produces a considerable accumulation of $OH^-$. According to recent reports, excessive assimilation of $NH_4^+$ by glutamine synthetase (GS) produces acid stress and has toxic effects on plants (*Hachiya et al., 2021*; *Witte, 2011*). To verify the relationship between acid stress of $NH_4^+$ assimilation by GS and plant salt sensitivity, we added GS inhibitor L-methionine sulfoximine (MSX) to a salt-stress medium to inhibit $NH_4^+$ assimilation (*Kiyomiya et al., 2001*; *Rawat Suman et al., 1999*) and found that MSX significantly inhibited the germination of WT seeds under salt stress, while WT germinated well in the control medium (*Figure 5a and b*). In addition, the percentage of seedlings with green cotyledons (*Figure 5c*) and root growth (*Figure 5d*) was severely restricted under salt-stress conditions. These results suggest that the acidification process of GS assimilation is beneficial to plant resistance to salt stress.

If the acidification process promotes plant resistance to salt stress, it follows that alkaline may exacerbate the inhibition of salt stress on seed germination and growth. Given that urea hydrolysis is an alkaline reaction process, we further conducted a thorough analysis of its impact on intracellular pH under salt stress. Initially, *Arabidopsis* seeds expressing a fluorescent pH indicator (PRpHluorin) were treated with NaCl and PPD (see 'Materials and methods'). To establish a correlation between PRpHluorin fluorescence ratios and pH values, an in vivo calibration was performed across the pH

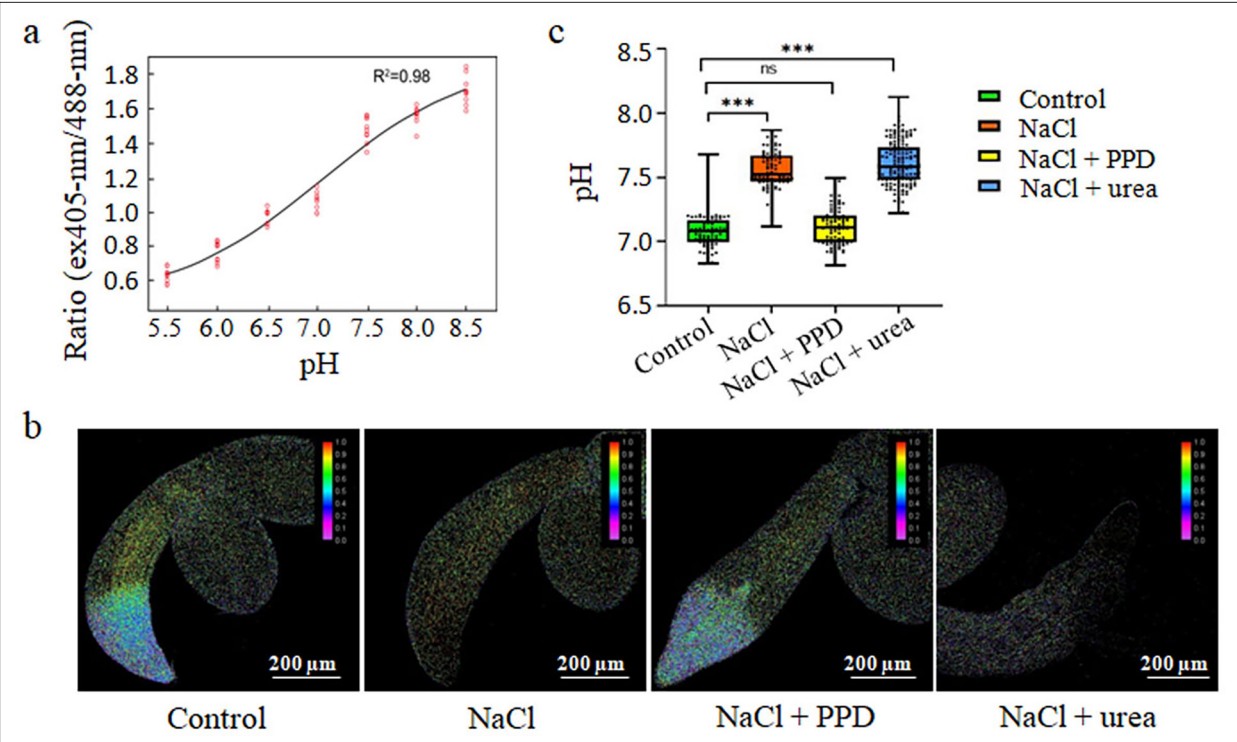

**Figure 6.** Inhibition of urea via arginine hydrolysis promotes seed germination under salt stress by lowering cell pH. (**a**) Calibration curve for pHluorin in seedlings adjusted to various pH levels using 50 mM MES-BTP (pH 5.2–6.4), 50 mM HEPES-BTP (pH 6.8–7.6), and 50 mM ammonium acetate. The curve plots the average fluorescence intensity ratios against the pH for 15 seedlings. (**b**) Fluorescence ratio images (emission 500–550 nm) of PRpHluorin expression seedlings grown in control, 135 mM NaCl medium, 135 mM NaCl with 15 µM PPD, and 135 mM NaCl with 10 mM urea, at 22°C for 3 days. Scale bar represents 200 µm. (**c**) Boxplots depict the cytoplasmic pH in root epidermal cells of seedlings, measured using the PRpHluorin fluorescence in the root elongation zone under treatments. Middle horizontal bars of boxplots represent the median, the bottom and top represent the 25th and 75th percentiles, and whiskers extend to the minimum and maximum. Statistical significance (***p<0.001) was revealed by the Student's *t*-test. The experiment was repeated more than three times with similar results.

The online version of this article includes the following figure supplement(s) for figure 6:

**Figure supplement 1.** Cytoplasmic pH of the root epidermal cells measurement.

range indicated in *Figure 6a*. We reasoned that in vivo calibration ensured that obtained pH ratios accurately reflected the intracellular environment, facilitating robust measurements of cytoplasmic pH over the pH range of 6–8. Then, we qualitatively observed panoramic pH in germinating seeds revealing a trend toward alkalinization in NaCl-treated seeds compared to the control (no salt treatment), especially evident in the epidermal cells of the root elongation zone (*Figure 6b*). However, the addition of PPD effectively alleviated this alkalinization of the cells under salt treatment conditions, restoring their pH to levels similar to the control.

Subsequently, we quantified the pH of epidermal cells in the root elongation zone at the onset of seed germination (3 days old) for PRpHluorin seeds. The intracellular pH was calculated for multiple cells in the elongated root epidermal region of germinated seeds, with each group consisting of no fewer than 10 samples. The results showed that the average cytoplasmic pH of the control was 7.07, and this pH significantly increased by 0.76 units after salt treatment (Student's *t*-test, p<0.001). In contrast, salt treatment with PPD restored the pH similar to the level observed in the control (*Figure 6b*). Additionally, exogenous salt plus urea treatment led to a clear trend toward higher pH in root cells compared to the salt-only treatment (*Figure 6b and c*), further affirming that the increase in intracellular pH induced by urea hydrolysis inhibited seed germination. These results validated our initial hypothesis that blocking urea hydrolysis by PPD reduces intracellular pH, effectively alleviating SISG. Excessive hydrolysis of arginine-derived urea elevates the cytoplasmic pH of seed radical cells, triggering SISG and impeding seedling growth in *Arabidopsis thaliana*.

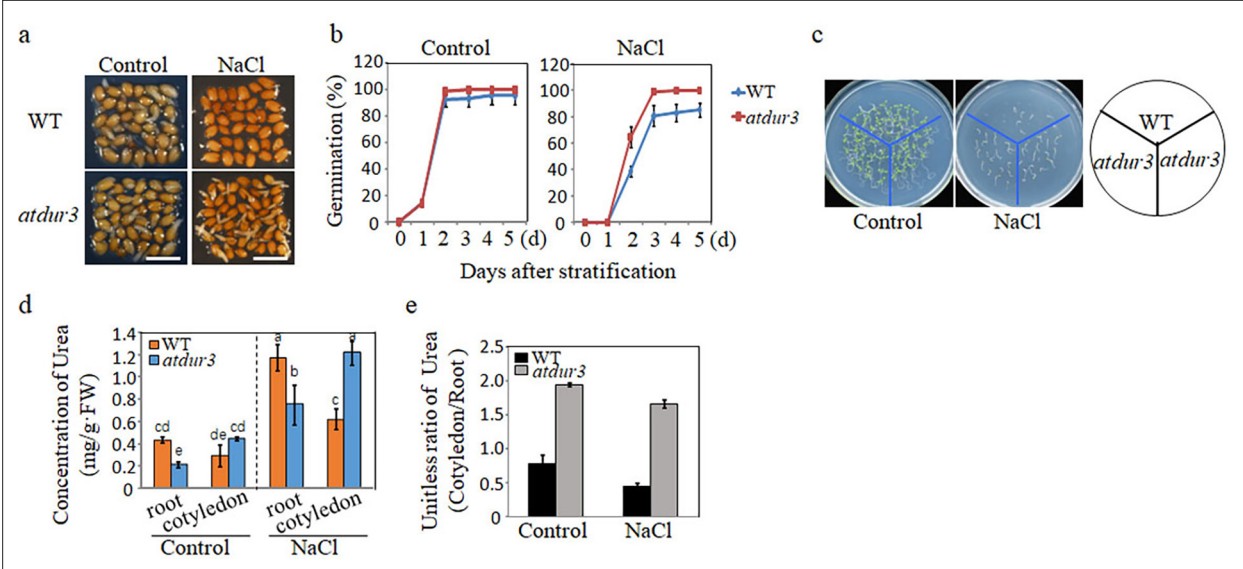

**Figure 7.** Arginine-derived urea transport and its role in salt-inhibited seed germination (SISG). (**a**) Seed germination phenotypes of WT and *atdur3* mutants were defined as the initial emergence of the radicle, which was observed and recorded at 48 hr of incubation following a 2-day stratification period at 4°C. Photographs captured under a stereomicroscope. Scale bar = 1 mm. (**b**) Germination rates of *atdur3* on ½ MS medium, containing 0 (control) or 135 mM NaCl, respectively, over time. (**c**) Growth comparison of WT and *atdur3* mutant seedlings on control and NaCl medium, photographed 10 days after germination. (**d**) Urea content analysis in roots and cotyledons of 5-day-old seedlings grown on control or 135 mM NaCl medium. (**e**) Calculation of urea (cotyledon/root) change ratio. In all experiments, seeds were freshly sowed and incubated under 16 hr light and 8 hr dark conditions at 22°C. The experiment was repeated three times; at least 30 seeds were counted in each replicate. The data was analyzed using a one-way ANOVA and Duncan's post hoc test, with different letters indicating significant differences (p<0.05).

The online version of this article includes the following source data and figure supplement(s) for figure 7:

**Figure supplement 1.** Characterization of *atdur3* mutant plants.

**Figure supplement 1—source data 1.** Complete original file of the full raw unedited *AtDur3* RT-PCR gel image.

**Figure supplement 1—source data 2.** Complete original file of the full raw unedited *Actin* gel image.

**Figure supplement 1—source data 3.** RT-PCR expression data of *AtDur3* in the WT and *atdur3* mutants.

## Blocking the transport of arginine-derived urea is beneficial for alleviating SISG

Based on our findings above, the actual trigger for SISG appears to be the increase in intracellular pH caused by the hydrolysis of urea in the roots, which is produced from arginine, rather than the effect of $NH_4^+$. It is noteworthy that urea in plants is mainly produced by the hydrolysis of arginine. During the mobilization of seed nitrogen stores, urea produced in the cotyledon is transported to the root by urea transporters. This raises the question: does blocking the transport of free urea generated by the arginine hydrolysis pathway also alleviate SISG? Moreover, urea transport might be involved in long-distance movement of urea from the cotyledon to the radicle, thereby affecting SISG. Exploring urea transport is essential to understand the SISG events caused by the arginine hydrolysis pathway during the mobilization of seed storage nitrogen at the tissue and organ level. To address these questions, here, we identified a loss-of-function mutant, *atdur3*, which lacks the urea transporter gene AtDur3 (*Figure 7—figure supplement 1*). The *atdur3* mutant line exhibited impaired growth on a medium containing urea as the sole nitrogen source (<5 mM) (data not shown). When *atdur3* and WT were germinated on control mediums, there was no significant difference in their seed germination ability and root development (*Figure 7a–c*). On the control medium supplemented with NaCl, the germination ability of WT seeds was significantly inhibited (the ability of radicle to break through the seed coat), and the germination rate of *atdur3* mutant seeds was significantly higher than that of WT (*Figure 7a and b*). These results suggest that blocking urea transport in vivo could effectively alleviate SISG. We hypothesized that blocking the transport of urea from arginine hydrolysis in the cotyledon to the root and blocking its hydrolysis in the root would promote the germination of seeds under salt stress. Therefore, we subsequently analyzed urea concentrations in the roots and cotyledon

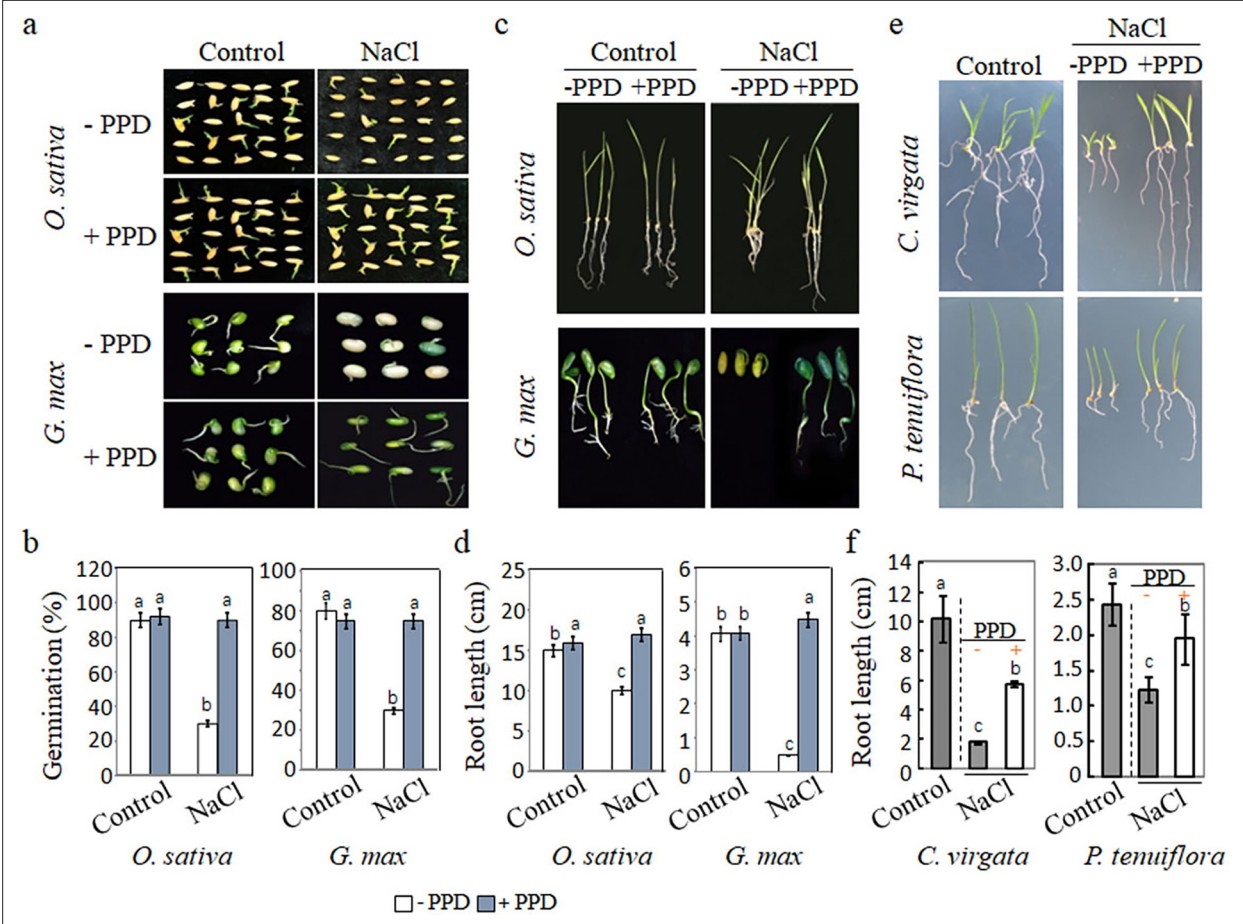

**Figure 8.** Effect of urea hydrolase inhibitor phenyl phosphorodiamidate (PPD) on the germination and seedling growth of *Oryza sativa* and *Glycine max*, *Chloris virgata* and *Puccinellia tenuiflora*. (**a**) Seeds of *O. sativa* and *G. max* were germinated on ½ MS medium (control; 0 mM NaCl) or ½ MS containing 150 mM NaCl with or without 15 µM PPD. Photographs were taken 14 days after germination. (**b**) Germination rates of *O. sativa* and *G. max* plants in control or 150 mM NaCl with or without 15 µM PPD were calculated after 5-day sowing. (**c**) Seedling growth of *O. sativa* and *G. max* treated with and without PPD under salt stress, photographed 14 days after germination. (**d**) Root lengths were determined 14 days after germination. (**e**) Seeds of *C. virgata* and *P. tenuiflora* were germinated 150 mM NaCl with or without 15 µM PPD compared to control (0 mM NaCl). Representative morphological images of the treated seedlings 14 days after germination. (**f**) Root lengths were determined 14 days after germination. The experiment was repeated three times with at least 30 plants per replicate. PPD, an inhibitor of urea hydrolase and a downstream metabolite of arginine.

of *atdur3* under salt stress. Under nonstressed normal conditions (control), root urea concentrations were lower in *atdur3* plants than in WT plants, but urea levels in cotyledons had no significant difference (*Figure 7d*). However, when plants were grown under NaCl stress, the urea levels were lower in *atdur3* roots than in WT roots. Still, urea levels were higher in *atdur3* cotyledons than in WT cotyledons (*Figure 7d*). We further analyzed the change ratio of urea (cotyledon/root), and the results show that urea was mainly accumulated in *atdur3* cotyledons under control or NaCl conditions (*Figure 7e*). These results further indicate that the transport of urea, the primary metabolite of arginine hydrolysis, plays an important role in SISG, that is, blocking the transport of arginine-derived urea from the cotyledon to the root is conducive to the alleviation of SISG.

## Effects of arginine hydrolysis pathway on salt tolerance in other plants

Taken together, these results underscore the pivotal role of arginase-dependent arginine catabolic pathways in alleviating SISG in *Arabidopsis*. To investigate the conservation of this role across plants, we examined two major crops, *Oryza sativa* and *Glycine max*, along with two halophytes, *Chloris virgata* and *Puccinellia tenuiflora*. The seeds of *O. sativa* and *G. max* exhibited robust germination in the control medium with or without PPD. However, germination of seeds was notably inhibited in salt medium, a hindrance significantly improved by additional of PPD (*Figure 8a and b*). Without PPD,

germination rate was 20–30% in the presence of NaCl, but with PPD supplementation, germination rates significantly increased to approximately 60–80% (*Figure 8a and b*). Furthermore, root lengths of the seedlings were significantly shorter in NaCl medium compared to control conditions, yet supplementation with PPD in the NaCl medium led to increased root lengths (*Figure 8c and d*). Similarly, studies on the typical halophytic plants demonstrated that under NaCl conditions treatment with PPD significantly reduced salt-stress-induced inhibition of shoot and root growth compared to untreated counterparts (*Figure 8e*). Notably, root lengths of samples treated with PPD under NaCl conditions were significantly higher than those without PPD treatment (*Figure 8f*). These observations collectively suggest that blocking arginase-dependent arginine catabolism can enhance salt-tolerant seed germination in both crops and halophytes, underscoring the universal effect of arginine catabolism on plant salt tolerance.

## Discussion

The loss of food due to farmland salinization is staggering, with an estimated economic impact of exceeding $27 billion annually (*Zörb et al., 2019*; *Kaleem et al., 2018*). Failure of seeds to germinate stands as a significant contributor to crop loss, particularly under salt stress conditions, which not only delays germination but also reduces the overall percent of successful germination (*Kojima et al., 2006*; *Kazachkova et al., 2016*). Therefore, developing effective strategies to mitigate the detrimental effects of salinity on seed germination and seedling establishment is imperative. Here we focused on the effect of seed reserve nitrogen mobilization on SISG. Arginase, a key enzyme in seed nitrogen storage mobilization, has garnered considerable attention in recent efforts aiming at understanding plant responses to abiotic stresses. Arginase-deficient mutants have been extensively utilized because of the competition among three enzymes, ARGAH, NOS, and ADC, for a common substrate, arginine. It has been proposed that blocking arginase activity indirectly enhances the NOS and ADC pathways, thereby increasing the plant defense response mediated by NO and PA (*She et al., 2017*; *Wang et al., 2011*; *Flores et al., 2008*). For example, ARGAH knockout mutants exhibit increased tolerance to various abiotic stresses, including water deficit, salt, and freezing, while lines overexpressing AtARGAH show decreased tolerance to these stresses (*Shi et al., 2013b*; *Flores et al., 2008*; *Wang et al., 2011*). The enhanced tolerance of AtARGAH knockouts to multiple abiotic stresses may arise from the competition for arginine between the ARGAH pathway and other metabolic pathways, such as those involving NOS and ADC. However, subsequent studies suggest that terrestrial plants might not possess animal-like NOS enzymes (*Zhao et al., 2015*; *Santolini et al., 2017*; *Jeandroz et al., 2016*). These studies showed that *Arabidopsis* nitric oxide synthase 1 (AtNOS1) was different from NOS, leading to its renaming as AtNOA1. Therefore, the link between the enhanced salt tolerance of *Arabidopsis* arginase mutants and NO accumulation via the NOS pathway is controversial. These explanations of ARGAH's role under salt stress are based solely on the competition of several metabolic pathways for substrate arginine and may be incomplete. Here we clearly indicated that by employing experiments involving arginase inhibitors, arginase gene mutants, as well as PPD and urease mutants, the arginine hydrolysis pathway (a two-step hydrolysis reaction) is directly involved in SISG (*Figure 1*). This finding not only challenges existing explanations but also underscores the significance of seed reserve nitrogen mobilization in the context of SISG.

How exactly does the arginine hydrolysis pathway trigger SISG? Our hypothetical model, summarized in *Figure 9*, elucidates the following process. The arginine hydrolysis pathway is implicated in triggering SISG through a series of interconnected physiological mechanisms. During seed germination, the activity of arginase and urease, especially particularly heightened under salt stress conditions, facilitates the conversion of arginine to urea and then into the ammonia pool (*Polacco et al., 2013*; *Siddappa and Marathe, 2020*; *Kojima et al., 2006*). Notably, arginase activity was significantly upregulated under salt stress (*Figure 2e*; *Polacco et al., 2013*; *Siddappa and Marathe, 2020*). This induction is further supported by strong evidence demonstrating the salt-induced expression of arginase and urease genes during seed germination (*Polacco et al., 2013*; *Siddappa and Marathe, 2020*; *Kazachkova et al., 2016*; *Lai et al., 2020*), and triggers significant accumulation of urea and $NH_4^+$ (*Figure 4a and b*). Together, these results indicate that the arginine hydrolysis pathway is induced by salt stress.

Interestingly, blocking the activity of arginase or urease necessarily triggers the accumulation of arginine or urea, respectively, yet paradoxically alleviates SISG. Thus, the toxic effects of arginine or

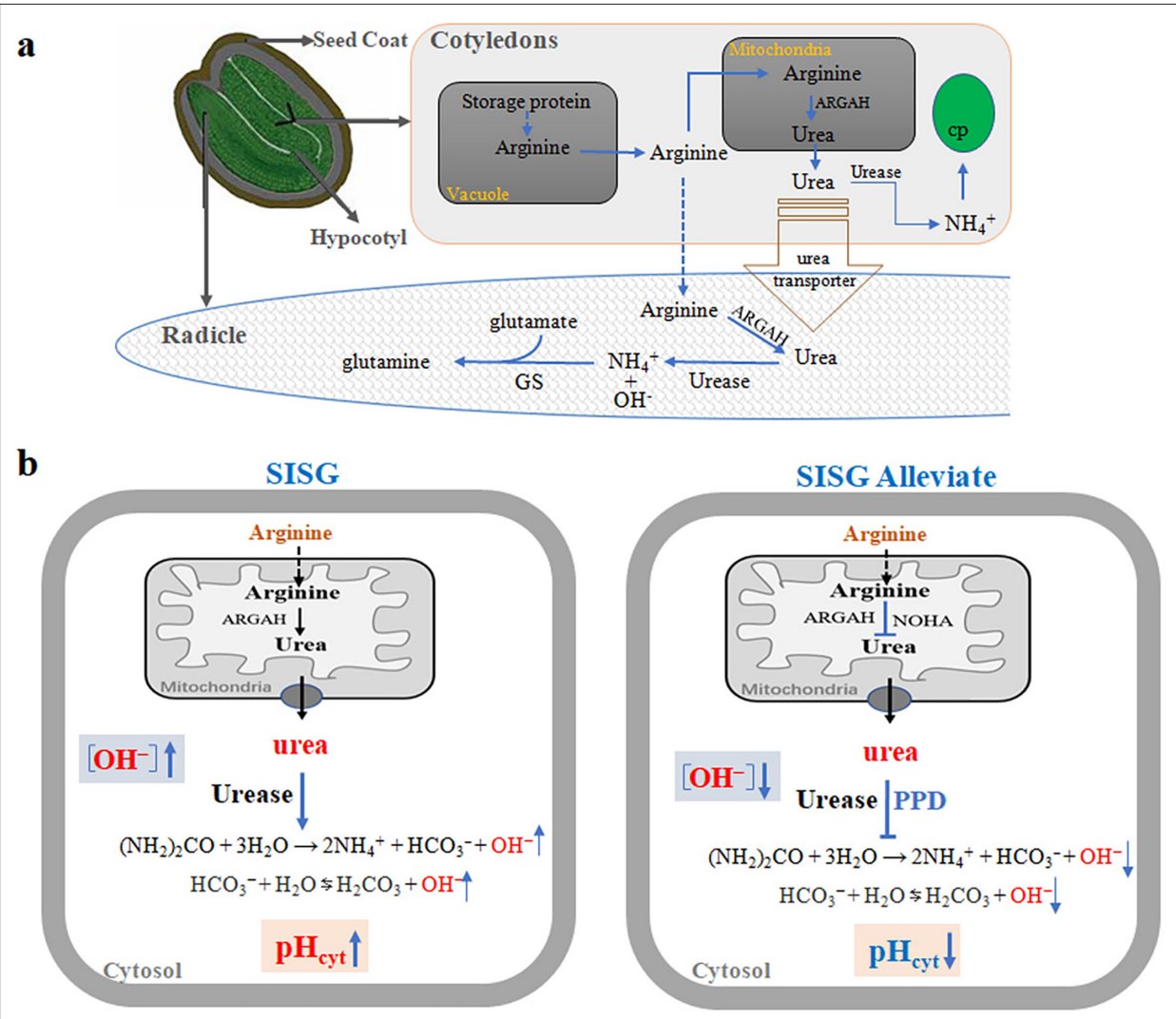

**Figure 9.** Hypothetical model for the regulation of seed germination by the arginine hydrolysis pathway under salt stress condition. (**a**) Salt stress highly induced the accumulation of arginine through the degradation of seed-stored proteins in the cotyledon. Then, arginine is hydrolyzed to urea through the action of arginine amidinohydrolase (ARGAH), and urea is further degraded by plant urease to form $NH_4^+$ and $OH^-$. (**b**) Salt stress stimulates the arginine hydrolysis pathway, leading to urea production by ARGAH, and its subsequent breakdown by urease, increasing $NH_4^+$, $HCO_3^-$, $OH^-$ levels, and cytoplasmic pH ($pH_{cyt}$), which inhibits seed germination (SISG). Application of arginine hydrolysis inhibitors (e.g., $N^G$-hydroxy-L-arginine [NOHA] for arginase, and phenyl phosphorodiamidate [PPD] for urease) or using *ARGAH* or *Urease* gene deletion mutants under salt stress can reduce $NH_4^+$, $HCO_3^-$, $OH^-$ levels, and decreased $pH_{cyt}$, thus promoting seed germination (SISG alleviate). The model delineates the urea hydrolysis process, resulting in increased cytoplasmic pH, $NH_4^+$, and alkalinity, contributing to the inhibition of seed germination under salt conditions, abbreviated as SISG (salt inhibits seed germination). PPD, an inhibitor of urea hydrolase.

urea accumulation can be excluded as a cause of SISG. Our results suggest that blocking arginase or urease activity leads to a decrease in $NH_4^+$ levels, which effectively mitigates SISG. This finding challenges our previous hypothesis that $NH_4^+$ accumulation due to urea hydrolysis under salt stress triggers SISG (*Bu et al., 2015*). However, our new results showed that the addition of ammonium to the medium under salt stress, on the contrary, effectively alleviated SISG (*Figure 4d*, *Figure 4—figure supplement 1*). This new unexpected experimental result refutes this seemingly plausible hypothesis that ammonium accumulation (ammonia toxicity) resulting from excessive urea hydrolysis is sufficient to trigger SISG.

Recent studies have shed light on the role of glutamine synthetase 2 (GLN2), primarily located in plastid, in inducing acid stress through assimilation of $NH_4^+$, which exerts toxic effects on plants (*Hachiya et al., 2021*). In addition, our results showed that MSX treatment fails to alleviate SISG

under salt stress, and interestingly, SISG shows a negative correlation with GS activity (*Figure 5*). This indicates that assimilation of $NH_4^+$ by GS may serve as an adaptive strategy for SISG. These results hint at the potential of cell acidification in alleviating SISG, which may explain why the addition of ammonium can effectively alleviate SISG. Traditionally, the hydrolysis of urea has been considered to produce ammonia and $CO_2$; in fact, the urea hydrolysis in the two-step hydrolysis reaction of the arginine hydrolysis pathway, '$(NH_2)_2CO + 3H_2O \rightarrow 2NH_4^+ + HCO_3^- + OH^-$', is indeed an alkalinization reaction. In addition, further hydrolysis of bicarbonate will also produce hydroxide '$HCO_3^- + H_2O \leftrightarrows H_2CO_3 + OH^-$', leading to further alkalization. Changes in cell parameters, including cytosolic pH (pHcyt), play a crucial role in both sensory and protective processes in plants under stress conditions. pHcyt acts as a pivotal regulator for numerous intracellular processes (*Felle, 2001*; *Gjetting et al., 2012*; *Shen et al., 2013*). In addition, to adapt to stressful environments, plants activate sensory and protective mechanisms within their roots, which are essential for maintaining the health of roots and shoots. It appears that root tissue plays a key role in plants' sensitivity to salinity (*Yang and Lee, 2023*; *Wu et al., 2018*; *Wu et al., 2015*). However, investigations focusing on salinity-induced, tissue-specific pH changes in roots or other tissues at salt concentrations have rarely been published, and they have mainly been conducted on *Arabidopsis* (*Ageyeva et al., 2022*; *Rombolá-Caldentey et al., 2023*). This scarcity is primarily due to the limitations of current measurement methods, which are not suitable for long-lasting observations on intact plants. Therefore, based on our findings on the influence of urea hydrolysis from the arginine storage in cotyledon on SISG and the associated alkalization process, we utilized transgenic *Arabidopsis* plants expressing fluorescent pH probes. These plants allowed us to assess the effects of salinity and the addition of PPD or urea on pHcyt in the cells of root elongation zones. Our investigation revealed that salt stress leads to an increase in cytoplasmic pH of seed radical cells, which results in SISG (*Figure 6*). Overall, these hypothetical models suggest that blocking arginase or urease activity under salt-stressed conditions can effectively mitigate the events leading to SISG, both depending on whether the alkaline hydrolysis reaction of the core metabolite urea occurs or is enhanced, providing new insights into SISG (*Figure 6*).

Salt inhibition of seed germination has been attributed to various signaling mediators and pathways (*Zörb et al., 2019*; *Kaleem et al., 2018*; *Zhu, 2002*; *Zhu, 2016*). $Na^+$ toxicity is thought to be one of the main factors triggering SISG, as evidenced by the mitigation of SISG in the salt-sensitive mutant *sos3* by blocking the arginine hydrolysis pathway. This result implies that initiation of the arginine hydrolysis pathway exerts an additional effect on SISG (*Figure 3*). Excessive hydrolysis of arginine leads to the accumulation of urea, and further hydrolysis of urea, coupled with elevated intracellular pH, leads to SISG events. Therefore, we posit that, in addition to $Na^+$ toxicity, SISG is also instigated by elevated intracellular pH resulting from urea hydrolysis. Salt sensitivity of *sos3* can be ameliorated by blocking urease activity and reducing intracellular pH under salt stress.

Urea is a plant metabolite derived from arginine hydrolysis or root uptake. In addition to the enzyme reaction required for urea hydrolysis, there are related urea transport steps in the mobilization process of nitrogen storage in seeds. In general, urea can be transported across plant cell membranes by high- or low-affinity transporters (*Polacco et al., 2013*; *Kojima et al., 2006*; *Bohner et al., 2015*). On the cellular and mitochondrial membranes, low-affinity urea transporters are mediated by PIP/TIP or TIP aqueous proteins, respectively (*Kojima et al., 2006*; *Bohner et al., 2015*). The Dur3 mediates the high-affinity transport of urea. The Dur3 mRNA level increased significantly during seed germination (*Bu et al., 2015*; *van Zelm et al., 2020*). According to our initial study, blocking the transport of free urea generated by the arginine hydrolysis pathway may also alleviate SISG (*Figure 7*), and AtDur3 may be involved in long-distance urea transport from cotyledons to radicles. Urea transport and the arginine hydrolysis pathway may synergistically affect SISG.

In summary, our study elucidates the hydrolysis reaction of arginine-derived urea in mobilizing stored nitrogen in seeds and its role in triggering SISG by raising the pH levels in seed radical cells. Although our evidence is primarily derived from studies on *Arabidopsis*, the application of PPD treatment across a diverse range of plants, including glycophytes, halophytes, graminaceous crops, and legumes, demonstrated consistent underlying principles. This consistency suggests that our findings may be applicable across the seed plant community (*Figure 8*), underscoring a broader relevance of our conclusions.

To conceptualize our findings, we summarized a hypothetical model illustrating the arginine hydrolysis pathway that regulates seed germination under salt stress (*Figure 9*). Salt stress significantly

induced the accumulation of arginine through the degradation of seed-stored proteins in the cotyledon. Then, arginine is hydrolyzed to urea by arginase, which is subsequently degraded by plant urease into $NH_4^+$ and $OH^-$ (*Figure 9a*). During this process, the inhibition of salt stress on seed germination and seedling growth can be alleviated by blocking either step of the arginine hydrolysis pathway (*Figure 9b*), with the rise in intracellular pH due to urea hydrolysis in the roots, being the actual trigger for SISG, rather than the effects of $NH_4^+$. The persistence of the two-step hydrolysis of arginine through evolutionary history, despite its potential adverse effects on seed germination, suggests that arginase and urease activities may represent an adaptive response to environmental changes. This resilience highlights the complexity of plant adaptation and provides further insight into SISG. Future studies focused on the enzymes involved in arginine metabolism and the regulatory mechanisms governing the allocation of arginine-derived nitrogen to defend against stress will enhance our understanding of the essential intermediates in this process. Additionally, examining the developmental switches between nitrogen storage and remobilization could help improve crop cultivation conditions and practices. Such studies will not only be pivotal for optimizing crop cultivation conditions but also for mitigating the negative economic and environmental ramifications associated with excessive urea application in agriculture.

# Materials and methods

**Key resources table**

| Reagent type (species) or resource | Designation | Source or reference | Identifiers | Additional information |
|---|---|---|---|---|
| Genetic reagent (*Arabidopsis thaliana*) | *Arabidopsis thaliana* mutant (atargah1) | ABRC (Ohio State University) | SALK_057987 | |
| Recombinant DNA reagent | CR-PCR-AtArgAH1AtArgAH2 (plasmid) | This paper | | M2CRISPR vector |
| Sequence-based reagent | atargah1/atargah2 | This paper | PCR primers AtArgAH1- FW/ RV | ACATGGGTTTCATTATGAAC/ CACAAAAGACTAAATACATG |
| Sequence-based reagent | atargah1/atargah2 | This paper | PCR primers AtArgAH2- FW/ RV | CCTTGCGGTCCTTGCCAAC/ ATAAACAGAATCTTATTGAG |
| Commercial assay or kit | Arginase Assay Kit | Bioassay | DARG-048 | |
| Commercial assay or kit | Urea Assay Kit | Bioassay | DIUR-048 | |
| Chemical compound, drug | $N^G$-Hydroxy-L-arginine | Sigma-Aldrich | H7278 | |
| Chemical compound, drug | Phenyl phosphorodiamidate | Macklin | p858176 | |
| Chemical compound, drug | L-Methionine sulfoximine | Sigma-Aldrich | M5379 | |
| Software, algorithm | SPSS | SPSS | RRID:SCR_002865 | |

## Plant materials and growth conditions

All experiments were performed with *Arabidopsis thaliana* Columbia (Col) wild-type plants and mutants in the Col background. The T-DNA insertion line SALK_057987 (*atargah1*) was ordered from the ABRC (Ohio State University), and homozygous lines were confirmed by genome PCR using primers 057987-LP or 057987-RP and a T-DNA primer LBP (*Supplementary file 1*). Other lines, including SALK_038002 (*aturease*), SAIL118_C11 (*atargah2*), and *sos3*, have been previously described (*Sessions et al., 2002*; *Zhu et al., 1998*). The double mutants *sos3*/*aturease* was produced by crossing, whereas *atargah1*/*atargah2* was produced using CRISPR-Cas9. Homozygous plants were selected from T$_2$ populations, and T$_3$ plants or further generations were used for analysis.

Wild-type and mutant plants were cultivated concurrently, with seeds collected simultaneously. Seeds were sown on half-strength MS (½ MS) medium containing 0.8% (w/v) agar, stratified for 2 days in the dark at 4°C, and then transferred to a 16 hr light/8 hr dark photoperiod at 22°C for 14-day cultivation. Seedlings were then moved to soil and cultured in a greenhouse under a photoperiodic cycle of 16 hr light and 8 hr dark photoperiod at 22°C. Fully developed and ripened brown siliques were collected for analysis.

## Seed germination assay

Seeds stored for a period ranging from 2 weeks to 3 months at room temperature were used for germination test. The after-ripened seeds were sterilized using 75% (v/v) ethanol and 10% (v/v) NaClO for 1 min, followed by three times washes in sterile water. Subsequently, they were plated on sterile filter paper to air-dry. Following sterilization, the seeds were sown on solid medium consisting of ½ MS supplemented with varying concentrations of NaCl (0, 135 mM), NOHA (Sigma, Germany; 0–5 μM), PPD (Macklin, China; 0–7.5 μM), and MSX (Sigma, Germany; 0–3 μM). Salt concentrations were set at 0 mM for controls and 135 mM for treatments unless noted otherwise. To reduce variation in germination, we sorted the seeds with an 80-mesh sieve to remove smaller seeds and selected seeds between 250 and 300 μm in size. The seedlings were stratified for 2 days in the dark at 4°C and then transferred to a 16 hr light/8 hr dark photoperiod at 22°C. At least 30 seeds for each genotype were used in three biological replicates. The germination event was defined as the initial emergence of the radicle, which was observed and recorded at 48 hr of incubation following a 2-day stratification period at 4°C. The assessment of germination, cotyledon-greening test, and radical growth used in the current study was as previously described (*Bu et al., 2015*). Seed germination rates were assessed daily in triplicates, with each plate containing over 35 seeds.

Seeds of wild-type *O. sativa* (Dongnong 421), *G. max* (Heihe 742), *C. virgata* and *P. tenuiflora* were surface-sterilized in 1% NaClO solution for 10 min, followed by washing three times in sterilized distilled water. Then, 30 seeds were sown on ½ MS (i.e., half the concentration of regular MS) supplemented with NaCl (0 or 150 mM) and PPD (0 or 15 μM), then subjected to stress conditions at 28°C for 14 days. Germination was defined by an obvious emergence of the radicle through the seed coat, with counts taken 3–5 days post-sowing. The germination rate was calculated as follows: germination rate (%) = (number of germinated seeds/total number of seeds) × 100%. Root length and fresh weight were measured after 14 days of cultivation. Data were analyzed using three biological replications, and statistical significance was determined using Duncan's test.

## Constructs and plant transformation

CRISPR-Cas9 technology was employed to design specific sgRNAs targeting *Arabidopsis AtArgAH1* and *AtArgAH2*. The gRNA-U6 fragment, once amplified, was cloned into the M2CRISPR vector (14,847 bp). The resulting construct, named CR-PCR-*AtArgAH1AtArgAH2*, was then introduced into *Arabidopsis* Columbia wild-type plants using stable transformation with *Agrobacterium tumefaciens*. Genomic DNA was extracted from young leaves of transformed plants and amplified by PCR using primers flanking the target sites to confirm the introduction of mutations. The PCR products were sequenced to identify double mutants, *atargah1/atargah2*. The primer sequences were using primers *AtArgAH1*-FW/RV and *AtArgAH2*-FW/RV (*Supplementary file 1*).

## Measurement of arginase activity

To detect arginase activity, WT, mutant lines of *Arabidopsis* (*atargah1*, *atargah2*, and the double mutant *atargah1/atargah*, with deletions in *AtArgAH1* or *AtArgAH2*) were germinated for 3 days on media containing 0 or 135 mM NaCl, following a stratification period of 48 hr at 4°C in darkness. The arginase activity was measured using a commercial assay kit (Bioassay, USA) following the manufacturer's instructions. Briefly, approximately 0.1 g of tissue was homogenized in 1 mL of extraction buffer on ice. After centrifugation at 8000 × *g* for 10 min, the supernatant was collected and placed on ice. For the assay, 10 μL of substrate buffer was added to the samples, and 200 μL urea reagent was added to the control tubes, respectively. After through mixing, the samples were incubated at 25°C for 60 min. Arginase activity was quantified by measuring the absorbance of the supernatant at 430 nm.

## Extraction and quantification of urea

For urea extraction, 1 mL of 10 mM ice-cold formic acid with the addition of the urease activity inhibitor (PPD) were added to about 100 mg sample to avoid urea hydrolysis. Urea concentration was then quantified using a commercial assay kit (BioAssay Systems, USA) following the manufacturer's instructions. Each sample mixture was vigorously vortexed twice before centrifuged at 16,000 rpm for 15 min at 4°C. The supernatant was carefully transferred to a fresh tube for analysis. Next, 200 μL of the provided color development reagent was added to each sample in a microcentrifuge tube. Tubes

were incubated 20 min at room temperature, and optical density of each sample was measured at 520 nm to quantify urea concentration.

## Ammonium concentration assay

Cotyledons and roots of *Arabidopsis* plants were gently washed with ice-cold Milli-Q water, dried with tissue paper, and immediately frozen in liquid nitrogen. For extraction, 0.5 g of frozen tissue was combined with 5 mL ice-cold extraction medium, and a small amount of quartz sand in a chilled mortar. The sample was then ground to a fine powder with a chilled pestle. Determination of the concentration of ammonium in plant tissue was conducted as previously described (*Bu et al., 2015*). $NH_4^+$ concentration was determined colorimetrically at 640 nm.

## Confocal laser scanning microscopy

*Arabidopsis* seeds engineered to express PRpHluorin were germinated on solid MS at 22°C for 3 days to facilitate pH measurement studies. The PRpHluorin gene was amplified using the specific primers (PRpHluorin-F and PRpHluorin-R) and then cloned into pBI121 vectors, which were modified to include the UBQ10 promoter. These constructs were introduced into WT Columbia-0 (Col-0) *Arabidopsis* plants via *Agrobacterium*-mediated transformation, employing the floral dip method followed by selection on kanamycin. For pH measurement in seedlings, fluorescent images were captured using a Zeiss LSM 880 confocal laser scanning microscope equipped with a ×20 or ×63 objective. The imaging was conducted in a sequential line scanning mode with settings described previously (*Shen et al., 2013*). Briefly, the emission (500–550 nm) of pHluorin, triggered by sequential excitation with 488 and 405 nm lasers, was used to calculate the pH using the calibration curve. In vivo calibration was performed on a subset of the seedlings at each experiment's conclusion, where they were immersed in pH equilibration buffers for 15 min (*Krebs et al., 2010*). The buffers contained 50 mM MES-BTP (pH 5.2–6.4) or 50 mM HEPES-BTP (pH 6.8–7.6), supplemented with 50 mM ammonium acetate. The emission ratio was plotted against pH, and sigmoidal curves were fit to the data using a Boltzmann equation, as described previously (*Gao et al., 2004*; *Schulte et al., 2006*). For post-treatment pH calculation, 10 seedlings with 100 cells per root were analyzed. The selected area was represented by the yellow dotted boxes (*Figure 6—figure supplement 1*), and pH values were deduced from the in vitro calibration curves using ImageJ software. Statistical analyses were conducted using one-way ANOVA in GraphPad Prism. To visually represent the pH profile, the grayscale ratio images were converted to pseudocolored images using ImageJ. Statistical significance (\*\*\*p<0.001) was revealed by the Student's *t*-test. The experiment was repeated more than three times with similar results.

## Statistical analysis

For physiological and biochemical data, an ANOVA was performed to investigate whether there was a significant difference between the samples. If a significant difference was found, a Duncan's significant difference test was performed to determine the specific samples with significant differences.

## Acknowledgements

This work was supported by the Heilongjiang Province Government Postdoctoral Science Foundation (LBH-Q18008) awarded to Yuanyuan Bu. This was further supported by the Program for Changjiang Scholars and Innovative Research Team in University (no. IRT17R99) awarded to Shenkui Liu. The funders had no role in the study design. We are grateful to Professor Jinbo Shen (Zhejiang Agriculture and Forestry University, Lin'an, China) for providing PRpHluorin seeds and technical support for cytoplasmic pH measurement.

# Additional information

## Funding

| Funder | Grant reference number | Author |
|---|---|---|
| Heilongjiang Province Government Postdoctoral Science Foundation | LBH-Q18008 | Yuanyuan Bu |
| Program for Changjiang Scholars and Innovative Research Team in University | IRT17R99 | Shenkui Liu |

The funders had no role in study design, data collection and interpretation, or the decision to submit the work for publication.

## Author contributions

Yuanyuan Bu, Conceptualization, Supervision, Writing - original draft, Writing - review and editing; Xingye Dong, Rongrong Zhang, Xianglian Shen, Yan Liu, Shu Wang, Data curation, Investigation; Tetsuo Takano, Validation, Writing - review and editing; Shenkui Liu, Conceptualization, Supervision, Writing - review and editing

## Author ORCIDs

Yuanyuan Bu  https://orcid.org/0000-0002-7133-613X

## Decision letter and Author response

Decision letter https://doi.org/10.7554/eLife.96797.sa1
Author response https://doi.org/10.7554/eLife.96797.sa2

# Additional files

## Supplementary files

• Supplementary file 1. PCR primer sequences used in the current study. The T-DNA insertion homozygous lines SALK_057987 (*atargah1*) were amplified using primers 057987-LP or 057987-RP and a T-DNA primer LBP. The *AtDur3* gene was amplified using the specific primers AtDur3-FW and AtDur3-RV. The *Actin* gene was amplified using the primers AtActin-FW and AtActin-RV. The PCR products of double mutants *atargah1/atargah2* were sequenced using primers AtArgAH1-FW/RV and AtArgAH2-FW/RV. The PRpHluorin gene was amplified using the specific primers PRpHluorin-F and PRpHluorin-R.

• MDAR checklist

## Data availability

Figure 2 - supplement 1 - source data contain the sequences for double mutants argah1/argah2. Figure 7 - supplement 1 - source data contain the original files of the full raw unedited gels of gene expression.

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
