## [Editor Report]

This important study advances our understanding of the molecular mechanism underlying the inhibition of seed germination and seedling growth by salt stress. The evidence supporting the conclusions is convincing, with rigorous genetic, physiological, and metabolic analyses. This paper will be of interest to plant biologists and crop breeders.

---

## [Decision Letter]

**Decision letter after peer review:**

Thank you for submitting your article "Unraveling the Role of Urea Hydrolysis in Salt Stress Response during Seed Germination and Seedling Growth in *Arabidopsis thaliana*" for consideration by *eLife*. Your article has been reviewed by 3 peer reviewers, one of whom is a member of our Board of Reviewing Editors, and the evaluation has been overseen by Jürgen Kleine-Vehn as the Senior Editor.

Essential revisions (for the authors):

1) If urea hydrolysis is causing the cytosolic pH to increase only in the roots, is important to confirm that is only happening there or more in the roots than in other organs.

2) Where is this urea coming from? Is this coming from the shoot? Would that transport be necessary for SISG?

3) It is worth evaluating the effects of different pHs from acid to alkaline on SISG.

4) A higher salt concentration (and at their optimum temperature) is necessary to confirm if the mechanism is valid for other species, especially for the halophytic plants

More details are provided below

*Reviewer #1 (Recommendations for the authors):*

Suggestions to improve.

The main idea of the first paragraph in the introduction is not clear (it's confusing whether you will study urea or salt toxicity). First, you talk about urea, then urea toxicity (it should be referenced) and then you mention extreme soils. I think you should rewrite this paragraph with one main idea in mind. Maybe just keep the importance of the urea as an N source, and then link it with arginine metabolism. Urea toxicity and extreme soils could be introduced later (e.g. before the last paragraph of the introduction).

Previous knowledge about urea toxicity under salinity needs to be briefly described, at least Bu et al., 2015 findings.

pH reduction by salt stress. I suggest including a pharmacological approach to reduce (and increase) pH or scavenge OH to confirm that the cause is the change in the pH and no other mechanism.

Urea transport. Are you hypothesizing that urea accumulating in the roots is coming from leaves? Would that be necessary for SISG? I think these results expand the knowledge about SISG but they sound preliminary. I think at least you may need to connect these findings with the previous results, analyzing the atdur3 mutants' germination under salt stress in combination with PPD.

In L. 298-302 you suggest that NO and PA production by NOS and ADC pathways are not involved in the SISG responses you studied here. I'd like to read a deeper discussion about this since you did not evaluate those pathways here (you said these explanations may be "inadequate", so I also suggest saying "incomplete", at least you demonstrate the opposite).

It makes no sense that root growth is affected in the same way in soybean and rice compared to Chloris and Puccinellia. Might this be due to the suboptimum temperature? Also, halophytes are supposed to be more salt tolerant than halophytes. Why did you use 150 mM to test both the glycophytic and halophytic species? How were Chloris and Puccinellia germination rates?

*Reviewer #3 (Recommendations for the authors):*

1) It is worthy to evaluate the effects of 1/2 MS media with different pH values from acid to alkaline on SISG.

2) The biological significance of the two-step catalyzed arginine catabolic pathway that is induced by salt stress and inhibits seed germination and seedling growth needs to be discussed.

---

## [Author Response]

Essential revisions (for the authors):1) If urea hydrolysis is causing the cytosolic pH to increase only in the roots, is important to confirm that is only happening there or more in the roots than in other organs.

Thank you for your insightful question. In our initial experimental design, we aimed to determine the pH levels in various parts of plants, including leaves, stems, and roots. However, we encountered significant challenges due to the limitations of our measurement technology. Specifically, the auto-fluorescence of chlorophyll and complex structures of leaves and other organs interfered with our ability to obtain accurate pH readings. This issue is common in pH measurement in other labs, as current literature often focuses on measuring pH in roots.

Given these constraints and the focus of our study on root physiology, we concentrated our efforts on analyzing the pH levels in fresh root tip tissue, where pH determination was more feasible and reliable. This decision aligns with our research objectives to explore the impact of urea hydrolysis on root function and soil-root interactions, which are critical for nutrient uptake and plant health.

We acknowledge the importance of understanding how these processes affect the entire plant, and we have discussed the limitations of our study and potential future research directions in this context (Lines 357-372). Future studies could leverage advancements in measurement technologies or innovative methodologies to overcome these challenges and provide a more comprehensive understanding of pH variations in different plant organs.

2) Where is this urea coming from? Is this coming from the shoot? Would that transport be necessary for SISG?

Thank you for your excellent question. Urea may be absorbed from the external environment, or produced from the degradation of cotyledon storage nitrogen. In this study context, we only considered the key role of endogenous urea hydrolysis from the degradation of cotyledon storage N during seed germination. We used arginase mutants and inhibitors to analyze the effect of urea hydrolysis on salt inhibition of seed germination. Furthermore, our experiment with urea transport mutants confirmed that blocking urea transport can alleviate salt inhibition of seed germination, which means that urea transport, potentially from the cotyledons to the germinating tissues, is necessary for salt inhibition of seed germination observed in seed germination.

3) It is worth evaluating the effects of different pHs from acid to alkaline on SISG.

Thank you for your insightful comment and suggestions. In this study, we primarily used genetic materials to explore how internal changes in pH, resulting from urea hydrolysis, influence SISG. Your suggestion to examine the effects of varying exogenous pH levels, from acidic to alkaline, on SISG is indeed a valuable research direction. We have initiated investigations in this domain and have observed some preliminary trends that hint at the complex role of pH in modulating SISG. These investigations are guided by the understanding that pH can significantly impact nutrient availability and enzyme activities critical for seed germination. We are employing a systematic approach to explore this, encompassing a broad spectrum of pH conditions. We anticipate that these findings will greatly enrich our understanding of pH regulation in SISG and look forward to discussing these in detail in our subsequent publication.

4) A higher salt concentration (and at their optimum temperature) is necessary to confirm if the mechanism is valid for other species, especially for the halophytic plants

We sincerely apologize for the oversight and appreciate your valuable feedback. It should be clarified that the processing temperature for our study subjects, such as rice, soybeans and other materials, is set at 28℃, reflecting the optimal temperature range for these species' germination and early growth stages. This has been corrected in the revised manuscript.

For the treatment of high salt concentration, we conducted a series of preliminary experiments with varying salt concentrations. We observed that the addition of PPD consistently alleviated the salt inhibition of seed germination across all tested concentrations. The chosen salt concentration for our main experiments was determined based on these pre-experimental results, aiming to illustrate a balance between ecological relevance and the clarity of the observed effects.

We acknowledge the reviewer's point on testing the mechanism in halophytic plants, which are inherently adapted to thrive in high salinity conditions, this indeed presents an intriguing direction for future research. Investigating the response of halophytes could provide valuable insights into the universality and limits of the observed mechanisms and is a direction we are considering for subsequent studies.

More details are provided belowReviewer #1 (Recommendations for the authors):Suggestions to improve.The main idea of the first paragraph in the introduction is not clear (it's confusing whether you will study urea or salt toxicity). First, you talk about urea, then urea toxicity (it should be referenced) and then you mention extreme soils. I think you should rewrite this paragraph with one main idea in mind. Maybe just keep the importance of the urea as an N source, and then link it with arginine metabolism. Urea toxicity and extreme soils could be introduced later (e.g. before the last paragraph of the introduction).Previous knowledge about urea toxicity under salinity needs to be briefly described, at least Bu et al., 2015 findings.

Thank you for your insightful comment and suggestions. In response, we have carefully revised the first paragraph of the introduction to focus more cohesively on urea's role as a vital nitrogen source in agricultural practices. We clarified the transition from discussing urea's importance to its potential toxicity and the challenges presented by extreme saline soils, ensuring a logical flow of ideas.

We have also made a concerted effort to link urea's role to arginine metabolism more explicitly, outlining how these aspects are crucial to understanding the broader context of our study. The sections on urea toxicity and the impact of saline soils have been repositioned to precede the concluding paragraphs of the introduction, aligning with your suggestion for a more coherent structure.

Additionally, we have integrated a concise overview of previous findings on urea toxicity under salinity, specifically referencing seminal works such as Bu et al., 2015, to provide a solid foundation for our study's premise. These revisions are reflected in lines 57-71 for the restructured introduction and lines 103-107 for the detailed discussion on urea toxicity under saline conditions.

We believe these changes have significantly improved the clarity and flow of the introduction, accurately setting the stage for the subsequent sections of our manuscript.

pH reduction by salt stress. I suggest including a pharmacological approach to reduce (and increase) pH or scavenge OH to confirm that the cause is the change in the pH and no other mechanism.

Thank you for your valuable suggestion regarding the use of a pharmacological approach to manipulate pH levels. We recognize the potential of such methods to provide clearer insights into the role of pH changes in seed germination under salt stress. In this study, we primarily focused on understanding the endogenous pH changes resulting from urea hydrolysis stimulated by salt stress on seed germination, as this aligns closely with our research objectives of exploring internal physiological responses.

We acknowledge that exogenous acid treatment could offer a more direct method of manipulating pH levels. However, we were cautious of the potential for such treatments to introduce additional complexities, given their potential to interfere with other cellular and physiological processes, thereby complicating the interpretation of results related to seed germination.

Given these considerations, we did not pursue exogenous pH manipulation in this study. However, we fully appreciate the importance of this approach and are planning to incorporate pharmacological methods to both reduce and increase pH, as well as scavenge OH^-^, in our future work. This will allow us to more definitively determine the causal relationship between pH changes and seed germination under salt stress, and to distinguish these effects from other concurrent mechanisms.

Urea transport. Are you hypothesizing that urea accumulating in the roots is coming from leaves? Would that be necessary for SISG? I think these results expand the knowledge about SISG but they sound preliminary. I think at least you may need to connect these findings with the previous results, analyzing the atdur3 mutants' germination under salt stress in combination with PPD.

Thank you for your excellent question. Urea is not only absorbed from the external environment, but also be produced from the degradation of nitrogen stored in the cotyledon, which is thought to be a significant internal source of urea produced in seedlings. However, our study specifically focuses on urea derived from cotyledon-stored nitrogen due to its significant role in the early stages of seedling development, particularly in the context of SISG.

In this study, we analyzed the effect of urea hydrolysis on SISG using arginase mutants and inhibitors, and demonstrated that by nitrogen arginine hydrolysis the urea produced influences SISG significantly in cotyledon.

Urea transport from the cotyledon to the roots, mediated by specific transporters such as *atdur3*, is essential. Our findings using the *atdur3* mutant, which lacks efficient urea transport, suggest that this transport is a crucial factor in triggering SISG. Therefore, we consider urea transport to be a necessary condition for SISG.

Regarding the combined effects of the atdur3 mutation and PPD on salt stress, our preliminary experiments (referenced in the figure below) indicate that the *atdur3* mutant's growth under salt stress does not significantly differ with or without PPD. In contrast, the wild type exhibits significant changes, underscoring that disrupting urea transport enhances the mutant's tolerance to salt stress. This is true regardless of the presence of PPD, which further blocks urea hydrolysis, highlighting the pivotal role of urea transport in SISG.

**Author response image 1. sa2fig1:** Effect of AtUrease inhibitor PPD on growth of Wild-type (WT) and <italic>atdur3</italic>.(A-D) germinated on half-strength MS or 135 mM NaCl with or without 15 µM PPD. Representative images show the morphology of seedlings 14 d after 2 days stratification at 4℃. (E) Root length was determined 14 d after the stratification.

In L. 298-302 you suggest that NO and PA production by NOS and ADC pathways are not involved in the SISG responses you studied here. I'd like to read a deeper discussion about this since you did not evaluate those pathways here (you said these explanations may be "inadequate", so I also suggest saying "incomplete", at least you demonstrate the opposite).

Thank you for your suggestions. We have expanded the discussion to clarify the independence of NOS and ADC pathways from the arginase pathway in the revised manuscript. Please find our revisions in lines 307-319: "For example, ARGAH knockout mutants exhibit increased tolerance to various abiotic stresses, including water deficit, salt, and freezing, while lines overexpressing AtARGAH show decreased tolerance to these stresses (Shi et al., 2013; Flores et al., 2008; Wang et al., 2011). The enhanced tolerance of AtARGAH knockouts to multiple abiotic stresses may arise from the competition for arginine between the ARGAH pathway and other metabolic pathways, such as those involving NOS and ADC. However, subsequent studies suggest that terrestrial plants might not possess animal-like NOS enzymes (Zhao et al., 2015; Santolini et al., 2017; Jeandroz et al., 2016). These studies showed that Arabidopsis nitric oxide synthase 1 (AtNOS1) was different from NOS, leading to its renaming as AtNOA1. Therefore, the link between the enhanced salt tolerance of Arabidopsis arginase mutants and NO accumulation via the NOS pathway is controversial. These explanations of ARGAH's role under salt stress are based solely on competition of several metabolic pathways for substrate arginine may be incomplete."

It makes no sense that root growth is affected in the same way in soybean and rice compared to Chloris and Puccinellia. Might this be due to the suboptimum temperature? Also, halophytes are supposed to be more salt tolerant than halophytes. Why did you use 150 mM to test both the glycophytic and halophytic species? How were Chloris and Puccinellia germination rates?

Thank you for your questions. We acknowledge the discrepancy in the reported culture temperatures for germination; Rice and soybean were indeed cultured at 28℃, not 22℃ as erroneously mentioned, which has been corrected in the revised text.

Regarding the salt tolerance of *Chloris* and *Puccinellia*, it's recognized that these species are generally more salt-tolerant. In our preliminary experiments across a range of salt concentrations, we observed that the addition of PPD consistently mitigated the salt-induced inhibition of seed germination and growth in all tested species. This led us to select a salt concentration for our definitive experiments that showed clear growth inhibition yet significant recovery upon PPD treatment.

As for the germination rates of *Chloris* and *Puccinellia*, they followed a similar trend to that of rice and soybean, aligning with the overall findings of our study.

Reviewer #3 (Recommendations for the authors):1) It is worthy to evaluate the effects of 1/2 MS media with different pH values from acid to alkaline on SISG.

Thank you for your insightful comment and suggestions. In our current study, we have focused on utilizing genetic materials to investigate the impact of internal pH changes, resulting from urea hydrolysis, on salt inhibition of seed germination (SISG). This approach primarily considers the effect of internal acid-base balance on seed germination. We acknowledge that the application of exogenous acids or bases to adjust pH could introduce more complex effects on in vivo pH responses, which represent a significant and more intricate biological process.

The exploration of exogenous pH influences on SISG is indeed a valuable research direction. We are currently undertaking studies in this area to delve deeper into the regulatory role of pH on SISG. As such, we have not yet obtained results regarding the effects of exogenous acid treatments on modulating pH levels. We anticipate discussing these findings and their implications in detail in our forthcoming paper.

2) The biological significance of the two-step catalyzed arginine catabolic pathway that is induced by salt stress and inhibits seed germination and seedling growth needs to be discussed.

Thank you for your suggestions. We have discussed the biological significance of the two-step catalyzed arginine catabolic pathway that is induced by salt stress and inhibits seed germination and seedling growth. Please find our revisions in lines 414-420: "The persistence of the two-step hydrolysis of arginine through evolutionary history, despite its potential adverse effects on seed germination, suggests that arginase and urease activities may represent an adaptive response to environmental changes. This resilience highlights the complexity of plant adaptation and provides further insight into SISG. Future studies focused on the enzymes involved in arginine metabolism and the regulatory mechanisms governing the allocation of arginine-derived nitrogen to defend against stress will enhance our understanding of the essential intermediates in this process."